# pH-Responsive Graphene Oxide-Based 2D/3D Composite for Enhancing Anti-Corrosion Properties of Epoxy Coating

**DOI:** 10.3390/nano14040323

**Published:** 2024-02-06

**Authors:** Jian Wang, Yangyang Cao, Jieru Wan, Meng Zhang, Yunqiang Li, Yanli Wang, Dalei Song, Tao Zhang, Jun Wang

**Affiliations:** Key Laboratory of Superlight Materials and Surface Technology, Ministry of Education, College of Materials Science and Chemical Engineering, Harbin Engineering University, Harbin 150001, China; wjian@hrbeu.edu.cn (J.W.); cyyheu@hrbeu.edu.cn (Y.C.); wanjieru@hrbeu.edu.cn (J.W.); 2016101528@hrbeu.edu.cn (M.Z.); lyq17861821305@hrben.edu.cn (Y.L.); songdalei@hrbeu.edu.cn (D.S.); zhangtao@mail.neu.edu.cn (T.Z.); junwang@hrbeu.edu.cn (J.W.)

**Keywords:** ZIF–90, graphene oxide, self-healing, coating, corrosion protection

## Abstract

The functionalized graphene oxide (GO)-based composites as fillers added into organic coatings are desired for realizing the longstanding corrosion protection of carbon steel. Here, the pH-responsive two-dimensional/three-dimensional (2D/3D) GO-based composite (ZIF–90–AAP/GO) was developed by environmentally friendly corrosion inhibitor 4-aminoantipyrine (AAP) anchored on the in situ growth of zeolite imidazolate framework–90 (ZIF–90) on the GO surface (ZIF–90/GO) through the Schiff base reaction. The active filler (ZIF–90–AAP/GO) was incorporated into an epoxy coating (EP) to obtain a high-performance self-healing coating on the surface of carbon steel. ZIF–90–AAP can greatly improve dispersion and compatibility of GO in EP. The low-frequency impedance modulus of ZIF–90–AAP/GO–EP can still reach up to 1.35 × 10^10^ Ω⋅cm^2^ after 40 days, which is about three orders of magnitude higher than that of the EP containing GO (GO–EP) relying on its passive and active corrosion protection. Meanwhile, ZIF–90–AAP/GO–EP exhibits excellent self-healing performance. The self-healing rate of ZIF–90–AAP/GO changes from negative to positive after 24 h, which results from the effective corrosion inhibition activity of ZIF–90–AAP for carbon steel based on the pH-triggered controlled release of AAP. The developed pH-responsive 2D/3D GO-based composite coating is very attractive for the corrosion protection of carbon steel.

## 1. Introduction

Carbon steel has extensive applications in the transportation and construction industries and marine fields because of its high thermal stability and mechanical properties as well as affordable cost [1,2,3,4,5,6,7]. Nevertheless, carbon steel is susceptible to corrosion especially in marine environments. Therefore, various methods have been taken to safeguard steel against corrosion, including electroplating, micro-arc oxidation, thermal spraying and the use of organic coatings [8,9,10,11]. Among them, organic coatings are the most common anti-corrosion strategy. Organic coatings can effectively provide long-lasting protection for carbon steel by isolating the steel substrate from corrosive media [12]. Nevertheless, the micropores can arise from the solvent evaporation of pure organic coatings during the curing process, which unavoidably influence the barrier properties of the coatings [13].

One effective method that has been undertaken to improve the shielding performance of organic coatings is the incorporation of two-dimensional (2D) fillers such as graphene oxide (GO), basalt, molybdenum disulfide and boron nitride [14,15,16,17]. GO as a prevailing 2D material is widely used in materials science, biomedicine, environmental protection, energy sensor and so on, originating from its unique chemical and physical properties [18,19,20,21]. GO also has attracted considerable attention in the field of corrosion protection of metal as a consequence of its high aspect ratio and mechanical stability, good electrical insulation and excellent impermeable properties [22,23]. Feng et al. prepared graphene oxide flakes using the chemical exfoliation method and incorporated them into epoxy resin to synthetize the GO/EP coating. The GO flakes extended the penetration path of corrosive media to the substrate, thereby reinforcing the physical barrier properties of EP [14]. However, the corrosion protection capabilities of organic coatings only containing GO are still restricted for two reasons. Firstly, GO has weak dispersion in organic coatings due to its van der Waals force and π-π interactions [24]. Secondly, it lacks active corrosion protection, which leads to the reduction in the protection effect of coatings once the micro-defect appears [25,26,27].

The approach taken to overcome these problems is through introducing three-dimensional (3D) fillers loaded with corrosion inhibitors on the GO surface to achieve good dispersion and compatibility in the coating and enhance active anti-corrosion properties. The polymerized polydopamine nanospheres loaded with the corrosion inhibitor zinc ions onto the GO surface were prepared using a method inspired by mussels [28]. The benzotriazole-loaded halloysite nanotubes (HNTs) coated by polydopamine (PDA) were self-assembled on the GO surface [29]. Graphene oxide sheets were modified using benzotriazole-loaded titanium dioxide nanocapsules [30]. Furthermore, zeolite imidazolate frameworks (ZIFs), as a class of metal-organic frameworks, mainly consist of metal ions and organic ligands. ZIFs possess tunable chemical properties and highly ordered pore structures. ZIFs can be used as not only nanocontainers but also corrosion inhibitors. It is found that ZIF–67 could offer epoxy coating excellent active corrosion inhibition abilities based on its pH-responsive properties [31]. Moreover, ZIFs can significantly improve the compatibility of GO in organic coatings. Simultaneously, ZIFs can also endow the GO-based coating with good self-healing capabilities. For example, the pH-responsive epoxy-based anti-corrosion coating was prepared by a polyaniline-grafted GO nano-platform decorated by ZIF–9 [32]. The epoxy coating embedded with 2-mercaptobenzimidazole-inbuilt ZIF–8 modified GO nanosheets exhibited remarkable self-healing performance through pH-stimuli triggers [25]. Hence, organic coatings containing the ZIFs/GO composite loaded with corrosion inhibitors will have great potential application in the realm of the corrosion protection of metal.

Herein, we developed the pH-responsive 2D/3D composite (ZIF–90–AAP/GO) based on eco-friendly corrosion inhibitor 4-aminoantipyrine (AAP) anchored on ZIF–90 grown in situ on GO (ZIF–90/GO) via the Schiff base reaction. ZIF–90–AAP/GO was embedded into the epoxy coating to construct the ZIF–90–AAP/GO composite coating (ZIF–90–AAP/GO–EP) in order to strengthen the passive and active corrosion protection. The morphology and structure of ZIF–90–AAP/GO were characterized by transmission electron microscopy (TEM) and scanning electron microscopy (SEM), X-ray diffraction (XRD), Fourier transform infrared (FT–IR) and Nuclear Magnetic Resonance (NMR), respectively. Additionally, electrochemical impedance spectroscopy (EIS) and SEM-energy dispersive X-ray spectrometry (EDS) were utilized to evaluate the corrosion resistance and self-healing properties of ZIF–90–AAP/GO–EP. Finally, the anti-corrosion mechanism of ZIF–90–AAP/GO–EP for carbon steel was proposed.

## 2. Experimental Methods

### 2.1. Materials

2-Imidazolecarboxaldehyde (IC) was purchased from Shanghai Shaoyuan Chemical Technology Co., Ltd. (Shanghai, China). Zinc nitrate hexahydrate (Zn (NO_3_)_2_·6H_2_O) was bought from China National Pharmaceutical Holdings Chemical Reagent Co., Ltd. (Shanghai, China). 4-Aminoantipyrine (AAP) was purchased from Shanghai Yuanye Biotechnology Co., Ltd. (Shanghai, China). Q235 steel plate with the size of 60 × 35 × 3 mm was bought from Guangdong Jilong Metal Materials Co., Ltd. (Foshan, China). N-N dimethylformamide (DMF), methanol (MeOH), ethanol (EtOH), n-butanol (n-BuOH), triethylamine (TEA), potassium permanganate (KMnO_4_) and hydrogen peroxide (H_2_O_2_) were purchased from Tianjin Fuyu Fine Chemical Co., Ltd. (Tianjin, China). Natural graphite was bought from Nanjing Pioneer Nanomaterials Technology Co., Ltd. (Nanjing, China). Sulfuric acid and hydrochloric acid (HCl) were purchased from Harbin Reagent Chemical Factory (Harbin, China). Sodium nitrate (NaNO_3_) was bought from Shanghai Aladdin Co., Ltd. (Shanghai, China). Polyurethane-modified epoxy resin (EP) was purchased from China Qingdao Ocean New Material Technology Co., Ltd. (Qingdao, China).

### 2.2. Synthesis of ZIF–90

Typically, 4.8 mmol of IC was dissolved in 10 mL of n-BuOH and heated to 60 °C until it was fully dissolved. TEA (670 μL) was added to the above solution, cooled again to room temperature, and stirred for 5 min to get solution A. Then, Zn (NO_3_)_2_·6H_2_O (1.2 mmol) dissolved in 10 mL of n-BuOH was added to solution A. The resulting mixture was stirred under reflux at 75 °C for 10 min, centrifuged (10,000 rpm), rinsed with MeOH solution and dried overnight at 80 °C. Finally, ZIF–90 was obtained.

### 2.3. Synthesis of GO

GO was prepared using a modified Hummers process. First, NaNO_3_ was added to 70 mL of sulfuric acid under high-speed stirring in an ice-water bath. Then, 1 g of natural graphite and 3 g of KMnO_4_ were slowly added in turn under vigorous stirring and stirred for 1 h. Then, they were again stirred for 3 h at 35 °C and diluted to 500 mL by water. Finally, H_2_O_2_ was added. The mixture was centrifugated, washed with HCl solution and deionized water, and dried at −80 °C to ultimately obtain GO.

### 2.4. Synthesis of ZIF–90/GO

Typically, 0.1 g of GO was introduced into 10 mL of n-BuOH and ultrasonicated for 2 h. Next, 1.2 mmol of Zn (NO_3_)_2_·6H_2_O was added and dynamically agitated for 1 h to ensure that the zinc ions could anchor on the oxygen functional groups of GO. The anchored zinc ions furnished uniform nucleation sites for the growth of ZIF–90. Following this, 670 μL of TEA and 4.8 mmol of IC fully dissolved in 10 mL of n-butanol were added to the above solution to realize ZIF–90 in situ grown on the GO surface. The mixture was refluxed and stirred at 75 °C for 10 min. Finally, it was centrifugated, washed with MeOH and dried at 80 °C for 12 h to acquire ZIF–90/GO.

### 2.5. Preparation of ZIF–90–AAP/GO

The preparation of ZIF–90–AAP/GO is shown in Figure 1. ZIF–90/GO was incorporated into 25 mL of EtOH and sonicated to obtain solution A. Solution B was prepared by adding AAP to 25 mL of EtOH. Solution B was slowly dripped into solution A under stirring and refluxed for 6 h. The mixed solution was filtered, washed with MeOH and dried under vacuum overnight. Finally, the collected precipitate was ZIF–90–AAP/GO.

### 2.6. Preparation of ZIF–90–AAP/GO–EP

Firstly, Q235 carbon steel was polished by 400, 800, 1200 and 2000 mesh abrasive paper, then degreased with EtOH. Next, DMF solution of ZIF–90–AAP/GO was added to EP, which was coated to the pre-treated steel by the coating method to synthesize ZIF–90–AAP/GO–EP. The thickness of the prepared coatings was 40 ± 5 μm. Similarly, pure EP, EP containing GO coating (GO–EP), EP containing ZIF–90 (ZIF–90–EP) and EP containing ZIF–90/GO (ZIF–90/GO–EP) were also prepared using the above method, respectively.

### 2.7. Material Characterizations

The ZIF–90–AAP/GO morphology and structure were observed by SEM (TESCAN AMBER, Brno, South Moravia, Costa Rica) and TEM (Tecnai G2 S-Twin, Hillsboro, OR, USA). FT–IR was obtained by ATR–FT–IR (Nicolet iS20, Waltham, MA, USA). XRD was measured by Rigaku D/max-TTR-III (Tokyo, Honshu, Japan). NMR of ZIF–90–AAP/GO was analyzed using a fully automated NMR spectrometer (AVANQIII, Saarbrucken, Saarland, Germany). Raman spectra (Raman) were characterized by a confocal Raman microscope (WITec alpha300R, Ulm, Baden-Wuerttemberg, Germany). The release behaviors of AAP from ZIF–90–AAP/GO were examined by UV-visible spectroscopy (UV-vis, TU-1901, Beijing, China) spectrophotometer. EIS was performed using an electrochemical workstation (Autolab PGSTAT302 N, Utrecht, The Netherlands), which is a three-electrode system consisting of a working electrode (carbon steel with a test area of 3.14 cm^2^), a reference electrode (saturated calomel electrode) and a counter electrode (platinum sheet) with the amplitude sinusoidal voltage of 20 mV in the frequency range of 10^−2^–10^5^ Hz.

## 3. Results and Discussion

### 3.1. Characterization of ZIF–90–AAP/GO

TEM and SEM images of ZIF–90–AAP/GO are shown in Figure 2. GO exhibits silk-like texture with a few wrinkles and high transparency. ZIF–90 has a typical rhombic crystal structure with an average particle size of 400–500 nm. For ZIF–90/GO, ZIF–90 is densely distributed on the ZIF–90/GO surface, demonstrating that GO has no effect on the nucleation and growth of ZIF–90. For ZIF–90–AAP/GO, ZIF–90 retains its original crystal structure even after AAP is introduced. Meanwhile, for the particle size, ZIF–90–AAP is slightly larger than ZIF–90. Additionally, as shown in Figure 3, the average elemental N content of the ZIF–90/GO is 15.94%, whereas ZIF–90–AAP/GO is 22.89%, which correlates with AAP with nitrogen heterocycles anchored onto ZIF–90/GO.

The XRD patterns of various samples are shown in Figure 4a. For GO, the sharp peak at 9.7° is attributed to the characteristic diffraction peak of the GO (001) crystal plane [33]. The characteristic diffraction peaks of ZIF–90 can be observed at 7.3°, 14.5° and 17.9°, which are consistent with those of ZIF–90 reported in the literature [34]. For ZIF–90/GO, the diffraction peaks of ZIF–90 and GO can be detected. Meanwhile, the characteristic diffraction peak of GO becomes clearly weaker. These results show that ZIF–90 can grow in situ on the GO surface. For ZIF–90–AAP/GO, it shows the same diffraction peaks as ZIF–90/GO, suggesting that the added AAP hardly impacts on the crystal structure of ZIF–90. Nevertheless, the corresponding reduction in the peak intensity of ZIF–90 and GO can be obviously observed compared with ZIF–90/GO, which is related to the loading of AAP in ZIF–90–AAP.

Figure 4b shows the FT–IR spectra of different samples. For GO, the peak at 1052 cm^−1^ is assigned to the stretching vibration of C–O–C in the epoxy group on GO. The peaks at 1715 cm^−1^ and 3430 cm^−1^ correspond to the stretching of C=O in the carboxyl group and the hydroxyl group with O–H stretching vibrations, respectively [35]. For ZIF–90/GO, another peak at 1675 cm^−1^ corresponds to the stretching vibration of C=O of the IC ligand of ZIF–90 [36]. The absorption peaks at 1200–1500 cm^−1^ are ascribed to the stretching vibrational bands of the IC ligand on ZIF–90. The peak at 540 cm^−1^ is the Zn–N stretching vibration, implying the coordination of zinc ions with imidazole of IC [37,38,39]. All results obviously indicate that ZIF–90 is grown in situ on GO. In contrast, ZIF–90–AAP/GO shows a C=N stretching vibrational peak at 1630 cm^−1^, demonstrating that the aldehyde group on ZIF–90 can form Schiff base bonds with the amino group of AAP [40]. Furthermore, the two extra peaks at 1035 cm^−1^ and 875 cm^−1^ correspond to the in-plane bending vibration and out-of-plane bending vibration of the benzene ring on the loaded AAP [41]. The above results show that AAP can be anchored on ZIF–90/GO by the Schiff base reaction.

Figure 5a shows the 1H NMR spectrum of ZIF–90–AAP/GO. The signals that appear at 7–8 ppm belong to the protons of the aromatic ring [42]. Among them, the signal at 7.1 ppm is attributed to the imidazole ring of ZIF–90 [43]. Interestingly, there is a peak at 9.7 ppm, confirming the presence of the imine proton. This implies the formation of the Schiff base between the amino group in AAP and the aldehyde group in the IC ligand of ZIF–90 [42,44].

Figure 5b displays Raman spectra of various samples. For GO, two prominent characteristic peaks are observed at 1350 cm^−1^ (D peak) and 1600 cm^−1^ (G peak) [45]. For ZIF–90/GO and ZIF–90–AAP/GO, the G bands slightly shift to 1590 cm^−1^, respectively, implying the interaction between ZIF–90 and GO. In addition, the intensity ratio of the D band to G band (I_D_/I_G_) is generally applied to evaluate the disordered degree of GO [46]. The I_D_/I_G_ value of ZIF–90/GO (1.05) is higher than that of GO (0.91), indicating that the ZIF–90 growth on the GO surface increases the defective sites of GO. Moreover, the I_D_/I_G_ value of ZIF–90–AAP/GO (1.0) becomes lower compared with ZIF–90/GO after loading AAP, reflecting that AAP can compensate for the vacancy of GO [47].

Figure 5c illustrates the release behaviors of AAP in ZIF–90–AAP/GO at various pH conditions. The release of AAP from ZIF–90–AAP/GO is inhibited to a great degree under neutral (pH 7) and basic (pH 11) conditions. After 240 min, the total amount of released AAP is only 2.38 mg/L at pH 7 and 4.59 mg/L at pH 11. Unlike neutral and basic conditions, AAP can be quickly released in 15 min and reach the maximum amount (11.59 mg/L) in acidic conditions (pH 3), suggesting ZIF–90–AAP/GO possesses pH-responsive characteristics. In acidic conditions, disintegration of ZIF–90 is readily induced, and the Schiff base between AAP and ZIF–90/GO is also unstable [48]. These can cause the rapid release of AAP from ZIF–90–AAP/GO.

### 3.2. Dispersion and Compatibility of ZIF–90–AAP/GO in Epoxy Resin

The cross-sectional SEM images of various coatings are shown in Figure 6. EP presents a relatively smooth cross section without any fillers (Figure 6a). For GO–EP, its cross section becomes rough and exhibits an uneven distribution. These can be attributed to the fact that GO has poor dispersion in epoxy (Figure 6b, inset) resulting from the van der Waals force and π–π interactions of GO [24]. For ZIF–90–EP, it displays a relatively flat fracture cross section due to its homogeneous dispersion in epoxy (Figure 6c, inset). For ZIF–90/GO–EP, it shows a uniform scale-like cross section (Figure 6d), indicating that ZIF–90, as an active 3D MOF material, makes dispersion of GO good in the epoxy coating. Furthermore, for ZIF–90–AAP/GO–EP, the scale-like cross section becomes finer and more homogenous (Figure 6e), exhibiting the most compactness among the coatings. These indicate that AAP can further improve the dispersion and compatibility of ZIF–90/GO in EP, which will favor enhancing the barrier properties of the coating.

### 3.3. Anti-Corrosion Properties of Different Coatings

EIS is used to evaluate the long-term anti-corrosion properties of the different intact coatings (Figure 7) [47]. The impedance modulus at 0.01 Hz (|Z|_0.01Hz_) is generally indicative of the corrosion protection properties of the coating, so is the radius of the capacitive loop [49]. For EP, |Z|_0.01Hz_ dramatically declines from 8.44 × 10^9^ Ω⋅cm^2^ at the beginning of immersion to 3.38 × 10^6^ Ω⋅cm^2^ after 40 days, indicating that EP owns relatively poor barrier performance, owing to the rapid penetration of the electrolyte to the coating/substrate interface without fillers. For GO–EP, |Z|_0.01Hz_ gradually drops from initial 9.66 × 10^9^ Ω⋅cm^2^ to final 6.06 × 10^7^ Ω⋅cm^2^ over 40 days. On the one hand, |Z|_0.01Hz_ of GO–EP is much higher than that of EP in the later immersion, revealing that GO can boost barrier properties of EP due to its “labyrinth effect” from its two-dimensional nanosheet layer structure [50]. On the other hand, the continuous decline of |Z|_0.01Hz_ during the whole immersion can be related to poor dispersion of GO in EP originating from its strong van der Waals force [51]. Simultaneously, the capacitive arc radii of EP and GO–EP always shrink with immersion time, indicating their inadequate corrosion protection abilities due to the absence of active corrosion inhibitors. For ZIF–90–EP, |Z|_0.01Hz_ appears to fluctuate. Notably, there exists an apparent rising process between 10 and 20 days. This can correlate with the fact that ZIF–90–EP possesses certain self-repairing abilities. Interestingly, for ZIF–90/GO–EP and ZIF–90–AAP/GO–EP, |Z|_0.01Hz_ and the capacitive reactance arc radius basically has a similar trend with time over 40 days. There are at least two stages of first decreasing and then increasing. Furthermore, |Z|_0.01Hz_ can be further demonstrated; in Figure 8a, these two coatings exhibit much higher |Z|_0.01Hz_ than other coatings, indicating that they combine the barrier properties and the active corrosion inhibition. These greatly improve the corrosion protection for the substrate. Obviously, |Z|_0.01Hz_ of ZIF–90–AAP/GO–EP is greater than that of ZIF–90/GO–EP, suggesting that AAP can further endow the coating with extra self-healing properties [52].

The breakpoint frequency (F_b_) is the frequency corresponding to the phase angle of −45°. F_b_ is generally employed to evaluate the coating delamination degree. The lower F_b_ implies the smaller coating delamination [53]. As shown in Figure 8b, for EP, F_b_ continuously booms, implying the relatively large coating delamination, which can be resulted from the constant accumulation of corrosion products. For GO–EP, F_b_ slowly rises in the entire process, indicating that GO as the filler can prevent the electrolyte from penetrating to the substrate, thereby lowering the microscopic layered area of the coating. Additionally, for ZIF–90–EP, ZIF–90/GO–EP and ZIF–90–AAP/GO–EP, the growth of F_b_ from the initial to the final stage is extremely slow, especially for ZIF–90–AAP/GO–EP. These indicate that ZIF–90–AAP/GO–EP is hardly peeled from the substrate.

The equation reported by Ramamurthy was used to calculate the coating delamination index (D) to assess the peeling degree at the coating/substrate interface [54]. Equation (1) is as follows.
(1)$$D\left(\%\right)=100\times {\left(\frac{Z_{1}-Z_{2}}{Z_{1}}\right)}_{0.01\mathrm{Hz}}$$
where Z_1_ and Z_2_ are |Z|_0.01Hz_ at the beginning and the end of each stage, respectively. Figure 8c shows the curves of the D (%) value of each coating with the soaking time. The D values of all the coatings show a growing trend throughout the process, which is due to the gradual penetration of the electrolyte solution into the coatings. EP and GO–EP have the relatively high delamination index after 40 days, suggesting that they have poor protection for the steel substrate. ZIF–90–AAP/GO–EP shows a lower delamination index than EP and GO–EP after 40 days. This reveals that ZIF–90–AAP/GO can effectively hinder the penetration of the electrolyte into the EP matrix, thus minimizing the coating delamination and damage.

In addition, water absorption also can reflect the protective performance of the coatings [55]. It is determined by the following Brasher and Kingsbury Equation (2) [56].
(2)$$X_{v}\left(\%\right)=100\times \frac{\log\left(\frac{C_{c}\left(t\right)}{C_{c}\left(0\right)}\right)}{\log(80)}$$
where X_v_(%), C_c_(0) and C_c_(t) represent the volume fraction of water in the coating matrix, the coating capacitance at the initial time (t = 0) and time t, respectively. Figure 8d displays the evolution of the water uptake of various coatings with time. ZIF–90/GO–EP has a lower water adsorption rate compared with the above three coatings after 40 days, which results from the fact that the well-dispersed GO augments the barrier performance of the coating, and the imidazole groups on ZIF–90 also improve the crosslink density of the coating [31,57]. ZIF–90–AAP/GO–EP has the lowest water absorption rate after 40 days, indicating that ZIF–90–AAP/GO is able to significantly diminish the coating porosity, enhance the coating compaction and confer favorable water-proof qualities on the coating.

The equivalent circuit in Figure 9a is used for the EIS data fitting of the coatings except for EP and GO–EP after 20 days. R_s_, CPE_c_, R_c_, CPE_dl_, R_ct_ and W denote the solution resistance, coating constant phase element, coating resistance, double-layer constant phase element, charge transfer resistance and the Warburg impedance, respectively [58]. The evolution of R_c_ and R_ct_ with time is shown in Figure 10. ZIF–90–AAP/GO–EP has higher R_c_ and R_ct_ than other coatings after 40 days, which can be ascribed to the passive and active corrosion protection given by ZIF–90–AAP/GO.

### 3.4. Self-Healing Properties of Different Coatings

The self-healing properties of the different coatings are evaluated by the EIS measurement of the scratched EP (S–EP), ZIF–90/GO–EP (S–ZIF–90/GO–EP) and ZIF–90–AAP/GO–EP (S–ZIF–90–AAP/GO–EP) coatings (Figure 11). The capacitance arc radius can generally reflect the corrosion protection performance of the coating. For S–EP, the capacitance arc radius continuously decreases with time, and there is an extra inductive arc after 12 h, suggesting that EP only possess passive protection abilities. For S–ZIF–90/GO–EP, the capacitance arc radius first reduces in the first 36 h, then rises after 48 h and finally drops with time, implying that ZIF–90/GO–EP has a certain degree of self-repairing abilities. For S–ZIF–90–AAP/GO–EP, it is worth mentioning that it exhibits first decreasing and then rising trends two times throughout the immersion process. Furthermore, its |Z|_0.01Hz_ also has a similar change trend (Figure 12a). Meanwhile, it has the highest |Z|_0.01Hz_. These demonstrate that ZIF–90–AAP/GO–EP has excellent self-healing performance.

The self-healing rate is employed to further investigate the self-repairing performance of the coating [59]. Equation (3) is used as follows.
(3)$$r_{\mathrm{self}-\mathrm{healing}}=\frac{{\left|Z\right|}_{\mathrm{end}}-{\left|Z\right|}_{\mathrm{start}}}{t_{\mathrm{end}}-t_{\mathrm{start}}}\times 100\%$$
where |Z|_start_ and |Z|_end_ mean |Z|_0.01Hz_ at the beginning and the end of each immersion stage, respectively. As can be shown in Figure 12b–d, for S–EP, r_self-healing_ of all stages is always negative, indicating EP withstands severe corrosion without self-healing capabilities. Unlike EP, r_self-healing_ of ZIF–90/GO–EP varies from negative to positive in 36–48 h, suggesting that the self-healing process surpasses the corrosion process based on the fact that zinc ions and IC can be constantly released from ZIF–90/GO and absorbed on the surface of the substrate to heal the scratch. However, r_self-healing_ becomes negative in the later immersion stage (48–72 h), implying that ZIF–90/GO–EP only has the limited self-repairing effect. Interestingly, as for ZIF–90–AAP/GO–EP, r_self-healing_ still remains positive in the final stage (60–72 h), confirming that the continuous release of AAP from ZIF–90–AAP/GO can further enhance the self-healing performance of ZIF–90/GO.

The EIS data of S–EP, S–ZIF–90/GO–EP and S–ZIF–90–AAP/GO–EP after 72 h of immersion were analyzed using the equivalent circuit shown in Figure 9a. The resulting fitted parameters are presented in Appendix A. The inhibitory efficiency (IE) obtained from R_ct_ was calculated by Equation (4) [60,61] as follows:(4)$$\mathrm{IE}\left(\%\right)=\frac{R_{\mathrm{ct}\;2}-R_{\mathrm{ct}\;1}}{R_{\mathrm{ct}\;2}}\times 100$$
where $R_{\mathrm{ct}\;1}$ and $R_{\mathrm{ct}\;2}$ represent the charge transfer resistance values of S–EP and other scratched coatings (S–ZIF–90/GO–EP and S–ZIF–90–AAP/GO–EP), respectively. The fitted data and IE are included in Appendix A. The IE of S–ZIF–90–AAP/GO–EP reaches 89.37%, implying that ZIF–90–AAP/GO–EP possesses remarkable self-healing properties.

Figure 13 shows SEM, EDS and elemental mappings of the substrates of different scratched coatings. For S–EP, many corrosion products appear inside and around the scratch because EP is short on active protection properties. For S–ZIF–90/GO–EP, there are less corrosion products inside and around the scratch. Meanwhile, Zn and N elements can be observed, which can originate from the adsorption of IC released from ZIF–90/GO in the scratch and the disposition of zinc hydroxide generated by the interaction between the released zinc ions from ZIF–90/GO and hydroxide ions produced in the cathodic region. Furthermore, for S–ZIF–90–AAP/GO–EP, any corrosion products can hardly be observed inside and around the scratch, indicating that AAP can further enhance self-healing properties of ZIF–90/GO–EP. Additionally, it is known that iron oxides and/or hydroxides mainly constitute the corrosion products of the steel [62]. So, the O element content is indicative of the degree of the corrosion of the steel. Among the three scratched coatings, S–ZIF–90–AAP/GO–EP has the lowest O element content (0.49%). These reveal that ZIF–90–AAP/GO–EP possesses supreme active corrosion protection performance.

Appendix A displays the potentiodynamic polarization curves of the steel immersed in 0.35 wt% NaCl solution (Steel) with ZIF–90/GO or ZIF–90–AAP/GO (Steel–ZIF–90/GO or Steel–ZIF–90–AAP/GO) for 72 h. The related electrochemical parameters are shown in Appendix A. Meanwhile, the corrosion inhibition efficiency (η) obtained from the corrosion current density (i_corr_) was determined using the following formula (Equation (5)) [63].
(5)$$\eta =\frac{i_{\mathrm{corr}\left(0\right)}-i_{\mathrm{corr}}}{i_{\mathrm{corr}\left(0\right)}}$$
where i_corr(0)_ and i_corr_ denote i_corr_ of Steel and other steels (Steel–ZIF–90/GO or Steel–ZIF–90–AAP/GO), respectively. Obviously, Steel–NaCl/ZIF–90–AAP/GO possesses the highest corrosion inhibition efficiency and corrosion potential (E_corr_) and the smallest i_corr_. These reveal that the released inhibitors can effectively decelerate the corrosion rate of the steel.

### 3.5. Anti-Corrosion Mechanism of ZIF–90–AAP/GO–EP

Figure 14 represents the corrosion protection mechanism of ZIF–90–AAP/GO–EP.

(i)Passive corrosion protection mechanism

ZIF–90–AAP/GO as 2D/3D fillers can effectively enhance the passive corrosion resistance of EP. This is mainly attributed to the “labyrinth effect” provided by the 2D GO, which can lengthen the diffusion path of corrosive media to the coating/metal interface, thereby mitigating the corrosion of the substrate [64].

(ii)Active corrosion protection mechanism

The electrochemical process of carbon steel can be generally described as follows [25,65,66,67].

Anodic region:(6)$$\mathrm{Fe}\;\to {\mathrm{Fe}}^{2+}+e^{-}$$
(7)$${\mathrm{Fe}}^{2+}\to {\mathrm{Fe}}^{3+}+e^{-}$$
(8)$$2{\mathrm{Fe}}^{2+}+2H_{2}O+O_{2\;}\to \;2\mathrm{FeOOH}+{2H}^{-}$$

Cathodic region:(9)$$O_{2}+2H_{2}O+{4e}^{-}\to {4\mathrm{OH}}^{-}$$

It is obvious that corrosion causes the local pH to change. In the anodic region, the local pH drops. This acidic environment results in the breaking of the Schiff base bond between AAP and ZIF–90/GO [68,69,70,71], thus AAP is released from ZIF–90–AAP/GO, which has been confirmed by the UV–vis results. By virtue of its amino group and benzene ring, AAP can coordinate with ferrous ions to form the protective film to hinder the corrosion of steel. Meanwhile, the ligand IC of ZIF–90 can be protonated in the acidic conditions, which can accelerate the collapse of ZIF–90 [72,73,74,75], thereby releasing IC and Zn^2+^ from ZIF–90–AAP/GO. The released IC with the lone pair of electrons can react with iron ions with the empty d-orbitals. In addition, electrostatic attraction arises from the positively charged IC and few negatively charged Cl^−^ ions adsorbed on the substrate surface, which also can attenuate the corrosion [76]. In the cathodic region, local corrosion leads to the increase in pH. The released Zn^2+^ ions can interact with the generated hydroxide ions to form Zn (OH)_2_ deposited in the scratched area, which can curb the further corrosion of the substrate.
(10)$${\mathrm{Fe}}^{2+}+xH_{2}O+\mathrm{yAAP}+\mathrm{zC}l^{-}\to {\left[\mathrm{FeA}{\mathrm{AP}}_{y}{\left(\mathrm{OH}\right)}_{x}{\mathrm{Cl}}_{z}\right]}^{2-x-z}+{\mathrm{xH}}^{+}$$
(11)$${\mathrm{Fe}}^{2+}+xH_{2}O+\mathrm{yIC}+\mathrm{zC}l^{-}\to {\left[\mathrm{FeI}C_{y}{\left(\mathrm{OH}\right)}_{x}{\mathrm{Cl}}_{z}\right]}^{2-x-z}+{\mathrm{xH}}^{+}$$
(12)$${\mathrm{Zn}}^{2+}+2{\mathrm{OH}}^{-}\to \;\mathrm{Zn}{\left(\mathrm{OH}\right)}_{2}$$

Therefore, ZIF–90–AAP/GO–EP can provide carbon steel with long-term corrosion protection based on its excellent passive and active corrosion protection.

## 4. Conclusions

We developed pH-responsive 2D/3D ZIF–90–AAP/GO composite for enhancing corrosion-resistant epoxy composite coating by self-assembling ZIF–90 on GO nanosheets and grafting the corrosion inhibitor AAP on surface of ZIF–90 via the Schiff base reaction. ZIF–90–AAP/GO presents superior dispersion and compatibility in EP, which can be attributed to the fact that the stacking of GO nanosheets is suppressed because of the steric hindrance effect of ZIF–90–AAP grown on the GO surface. ZIF–90–AAP/GO–EP has the lower coating delamination index and water adsorption rate than the others after 40 days of immersion. Its |Z|_0.01Hz_ still reaches 1.35 × 10^10^ Ω⋅cm^2^ after 40 days. These show that ZIF–90–AAP/GO–EP can supply long-lasting efficient protection to the steel. Steel–NaCl/ZIF–90–AAP/GO has the highest inhibition efficiency, thereby effectively inhibiting the steel corrosion. Additionally, the EIS, SEM and EDS results show that ZIF–90–AAP/GO can greatly reinforce the self-healing performance of the GO-based epoxy coating based on the cooperative effects of controllably released AAP, IC and Zn^2+^ ions from ZIF–90–AAP/GO in response to pH stimuli. ZIF–90–AAP/GO–EP exhibits excellent passive barrier performance and active corrosion resistance.

## Figures and Tables

**Figure 1 nanomaterials-14-00323-f001:**
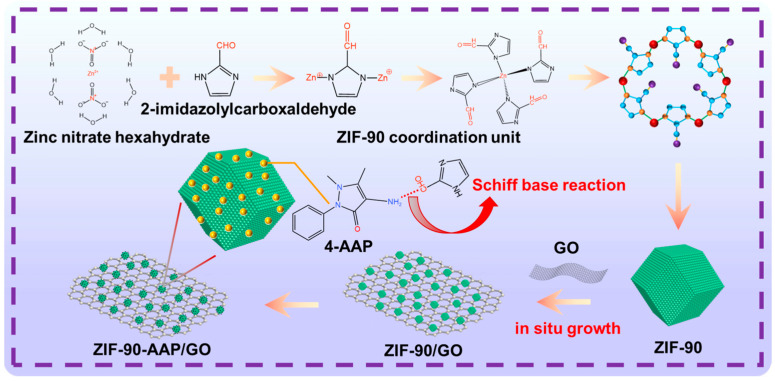
The schematic diagram of the preparation of ZIF–90–AAP/GO.

**Figure 2 nanomaterials-14-00323-f002:**
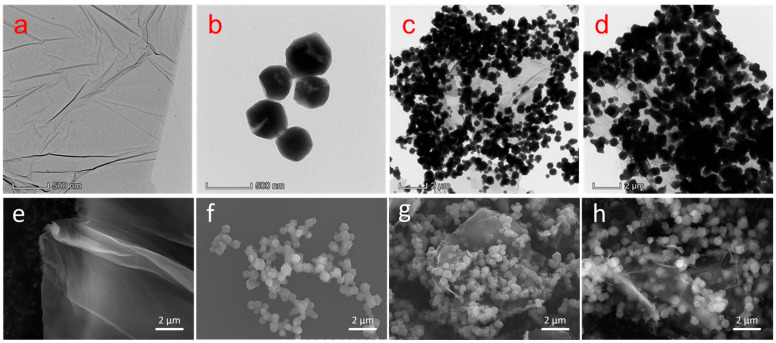
TEM (**a**–**d**) and SEM (**e**–**h**) images of GO (**a**,**e**), ZIF–90 (**b**,**f**), ZIF–90/GO (**c**,**g**), ZIF–90–AAP/GO (**d**,**h**).

**Figure 3 nanomaterials-14-00323-f003:**
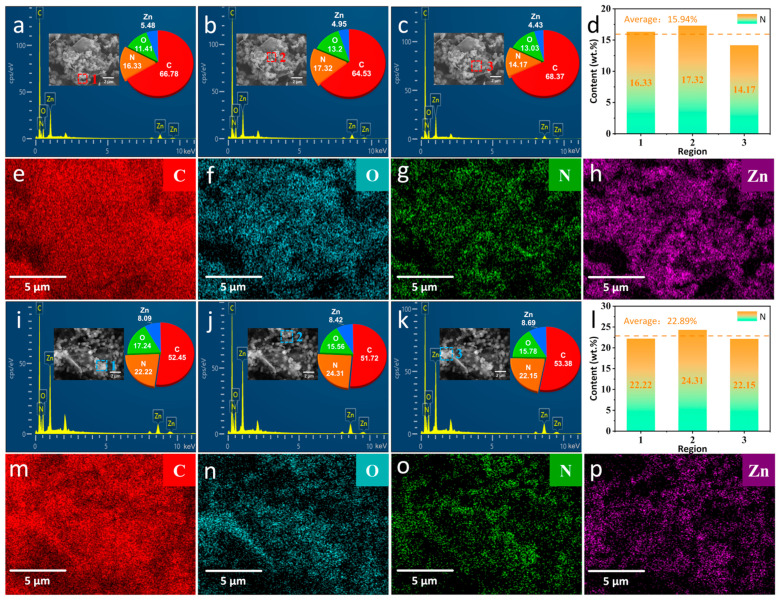
EDS and corresponding elemental mapping images of different regions of ZIF–90/GO (**a**–**h**) and ZIF–90–AAP/GO (**i**–**p**).

**Figure 4 nanomaterials-14-00323-f004:**
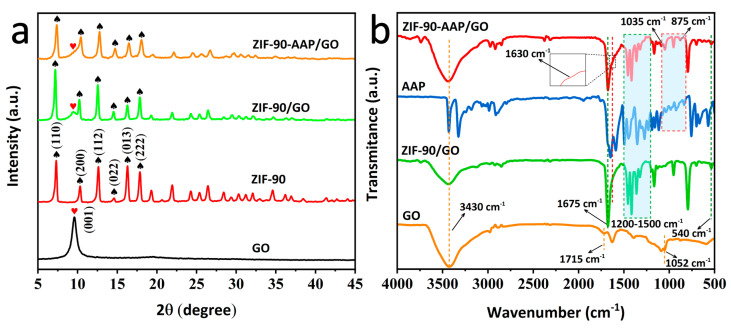
(**a**) XRD plots and (**b**) FT–IR spectra of the different samples. Note: characteristic diffraction peaks of GO (
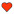
) and ZIF–90 (
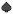
) are indicated.

**Figure 5 nanomaterials-14-00323-f005:**
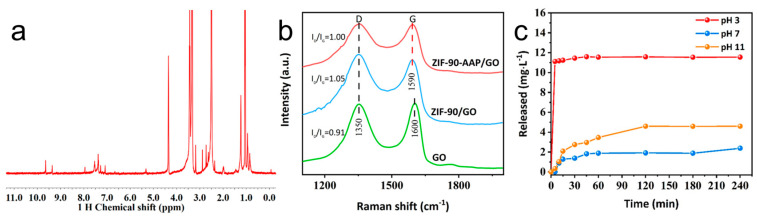
(**a**) The 1H NMR spectrum of ZIF–90–AAP/GO; (**b**) Raman spectra of different samples; (**c**) the release behaviors of AAP in ZIF–90–AAP/GO at different pH values.

**Figure 6 nanomaterials-14-00323-f006:**
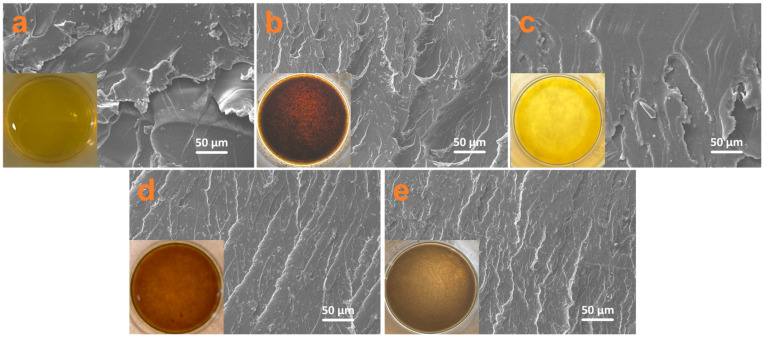
SEM images of the cross section for EP (**a**), GO–EP (**b**), ZIF–90–EP (**c**), ZIF–90/GO–EP (**d**) and ZIF–90–AAP/GO–EP (**e**). The inset is the optical image of pure epoxy (**a**) and different filler-added epoxy (**b**–**e**).

**Figure 7 nanomaterials-14-00323-f007:**
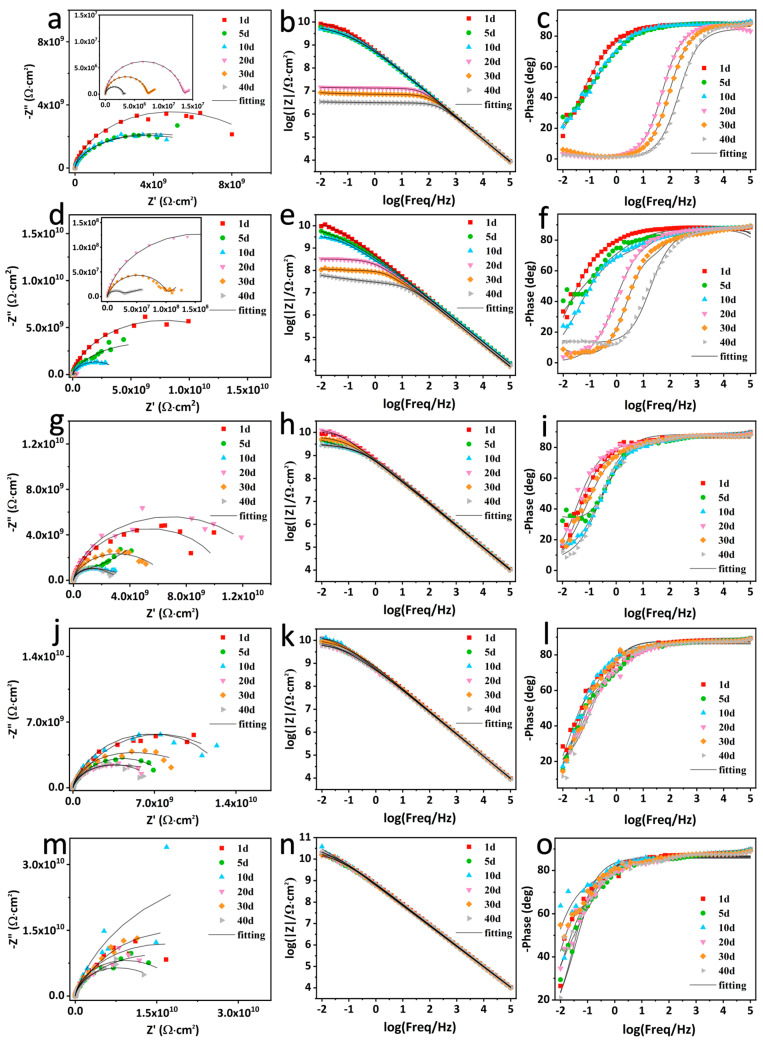
EIS plots of EP (**a**–**c**), GO–EP (**d**–**f**), ZIF–90–EP (**g**–**i**), ZIF–90/GO–EP (**j**–**l**) and ZIF–90–AAP/GO–EP (**m**–**o**) immersed in 3.5 wt% NaCl solution for 40 days.

**Figure 8 nanomaterials-14-00323-f008:**
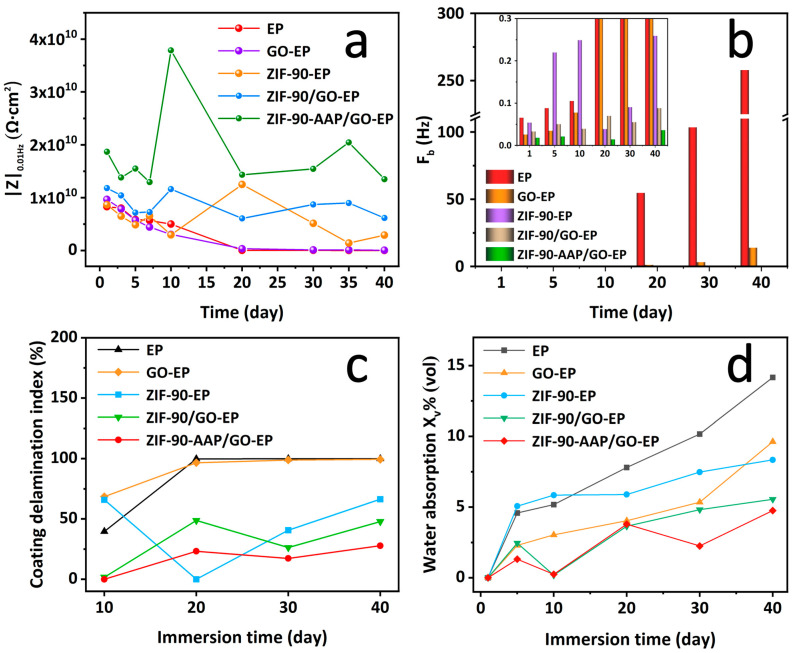
Evolution curves of (**a**) |Z|_0.01Hz_, (**b**) F_b_, (**c**) coating delamination index and (**d**) water absorption for different coatings with immersion time.

**Figure 9 nanomaterials-14-00323-f009:**
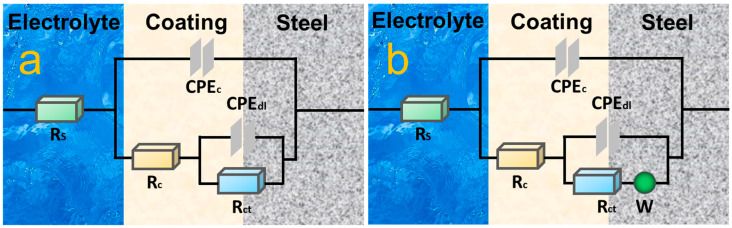
Equivalent circuits for fitting the EIS data of various coatings.

**Figure 10 nanomaterials-14-00323-f010:**
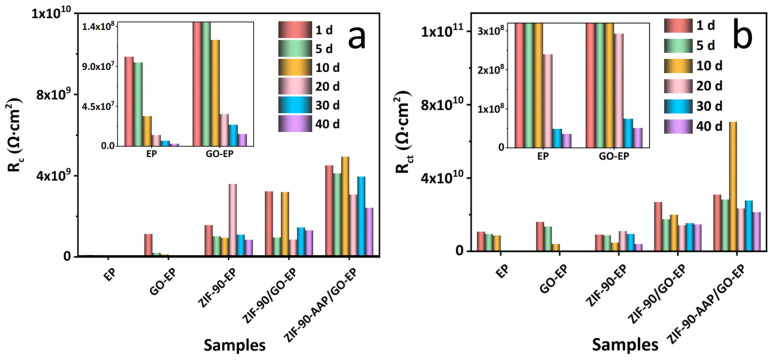
Evolution curves of (**a**) R_c_ and (**b**) R_ct_ of different coatings with time.

**Figure 11 nanomaterials-14-00323-f011:**
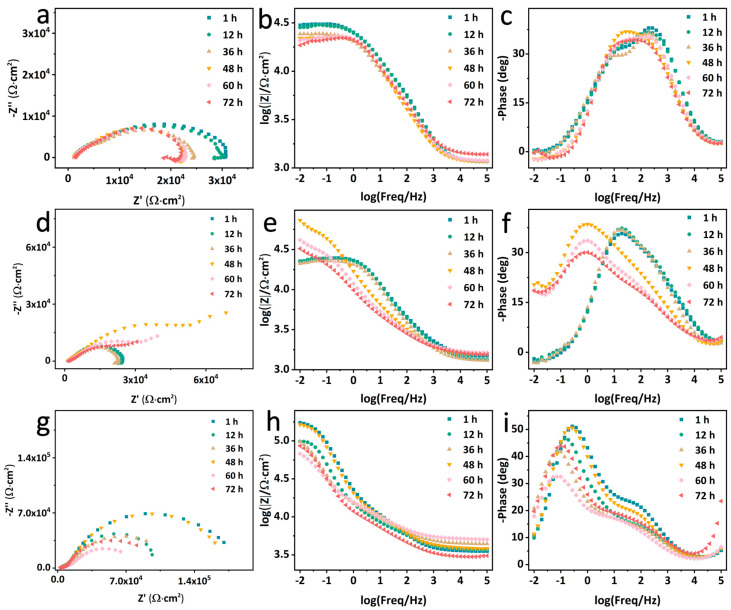
EIS plots of S–EP (**a**–**c**), S–ZIF–90/GO–EP (**d**–**f**) and S–ZIF–90–AAP/GO–EP (**g**–**i**) immersed in 0.35 wt% NaCl solution over time.

**Figure 12 nanomaterials-14-00323-f012:**
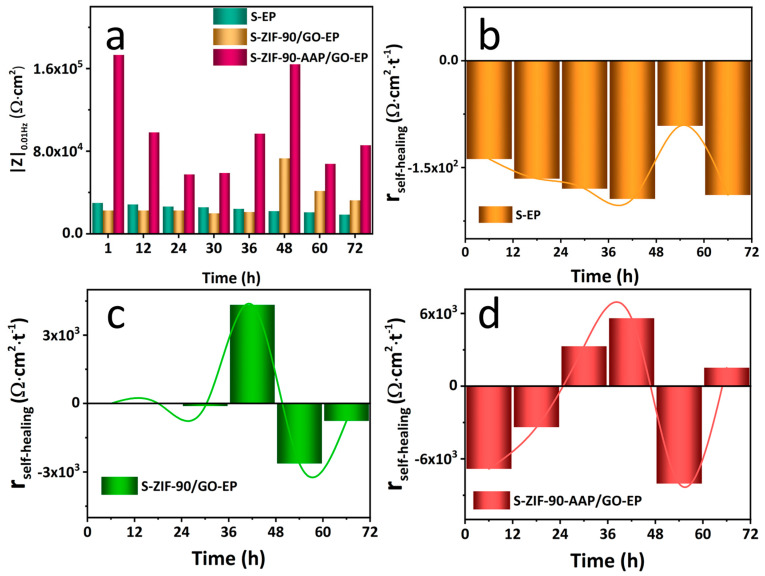
Evolution of (**a**) |Z|_0.01Hz_ and r_self-healing_ of different coatings with scratches (**b**–**d**) immersed in 0.35 wt% NaCl solution over time.

**Figure 13 nanomaterials-14-00323-f013:**
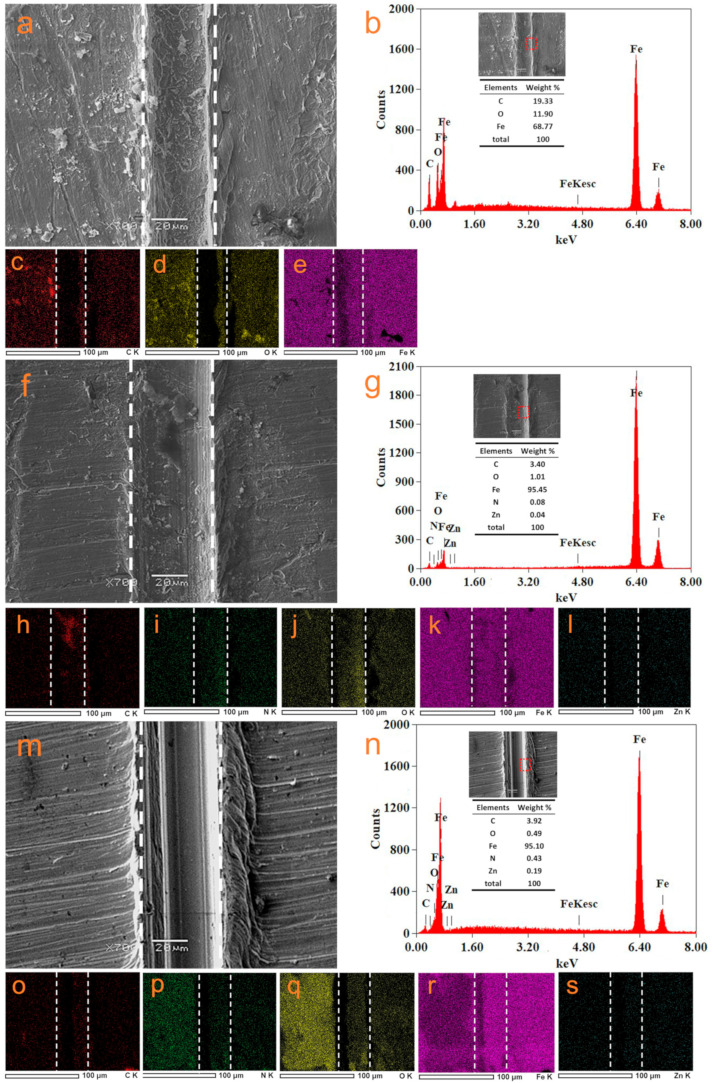
SEM (**a**,**f**,**m**) and EDS images (**b**,**g**,**n**) and the corresponding elemental mappings (**c**–**e**,**h**–**l**,**o**–**s**) of S–EP, S–ZIF–90/GO–EP and S–ZIF–90–AAP/GO–EP immersed in 0.35 wt% NaCl solution after 72 h.

**Figure 14 nanomaterials-14-00323-f014:**
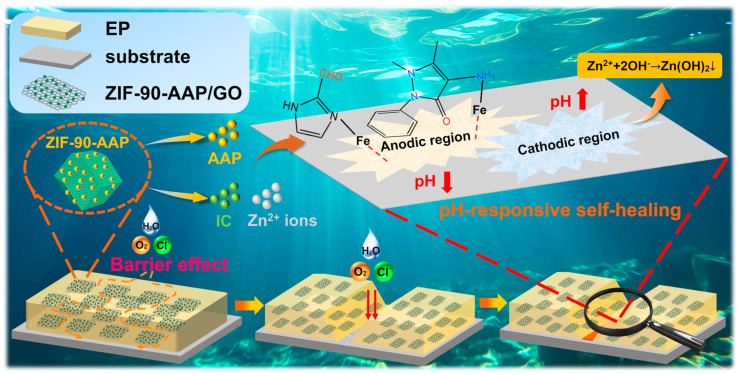
Schematic illustration of the anti-corrosion mechanism of ZIF–90–AAP/GO–EP.

## Data Availability

Data are contained within the article and supplementary materials.

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
