# Peer review of "pH-Responsive Graphene Oxide-Based 2D/3D Composite for Enhancing Anti-Corrosion Properties of Epoxy Coating"

_nanomaterials, 2024, doi:10.3390/nano14040323_

Round 1
Reviewer 1 Report
Comments and Suggestions for Authors
In this work, the authors have developed pH-sentitive GO composites for enhancing corrosion-resistant epoxy coating by self-assembling technique. This manuscipt can be more improved by addressing the questions below before get accepted.
1. Can the authors provide more details on the synthesis process of the ZIF-90-AAP/GO composite? Specifically, how do they ensure the uniformity of the composite and control the in-situ growth of ZIF-90 on the GO surface?
2. How does the performance of ZIF-90-AAP/GO-EP in anti-corrosion protection compare with existing corrosion inhibitors and coatings currently used in industry?
3. The manuscript mentions the passive and active corrosion protection offered by ZIF-90-AAP/GO-EP. Could the authors elaborate on the specific mechanisms by which this occurs?
4. Regarding the EIS results, can the authors discuss the significance of the three orders of magnitude difference in impedance modulus values between ZIF-90-AAP/GO-EP and GO-EP? How does this translate into practical terms for corrosion protection?
5. The study presents results up to 40 days. Are there any data or predictions about the long-term durability and effectiveness of the ZIF-90-AAP/GO-EP coating beyond this period?
6. While the corrosion inhibitor AAP is described as environmentally friendly, has the overall environmental impact of the ZIF-90-AAP/GO-EP composite been assessed, especially considering its full lifecycle?
Reviewer 2 Report
Comments and Suggestions for Authors
GO, 2D and others are mentioned in abbreviation several times. Please, correct it.
Replace materials and characterizations from Supplementary to main manuscript.
Figure 4: The peaks should be marked and described.
EIS description is very pure.
Considering the amount of experiments, the conclusions are too short.
Comments on the Quality of English Language
Language is understandable, but the style should be improved.
Reviewer 3 Report
Comments and Suggestions for Authors
The study shows a complete fabrication and characterization study of a pH-responsive corrosion inhibitor coating for carbon steel. The coating consists of a composite of graphene oxide and 4-aminoantipyrine anchored on zeolite imidazole. The weak part of this study is the corrosion inhibition characterization by electrochemical methods, which only provides impedance spectroscopy measurements and should ideally include potentiodynamic polarization.
Below some questions to be addressed:
1) When combining ZIF-90 with GO, the interlayer spacing increasing as observed by XRD diagrams. The reason behind should be clearly explained
2) Inhibition efficiency values for the different coatings should be provided following the definition of a reference article in the field, A. Kokajl et al., Simplistic correlations between molecular electronic properties and inhibition efficiencies: Do they really exist?, Corrosion Science 179, (2021) 108856. In particular, the authors can calculate the inhibition efficiency from charge transfer resistance values in the presence and absence of inhibitor layers. Also, it can be obtained from potentiodynamic polarization measurements, which should be described and included
3) The quality of Figure 7 is very poor and should be improved. In addition, the evaluation of anti-corrosion properties from diagrams displayed in this figure, is not properly explained and should be described in a much clearer way (first paragraph in section 3.4)
4) How were the scratches in the samples carried out? How can the authors discriminate the 'corrosion' products by SEM from stratch-induced products?
Comments on the Quality of English Language
Extensive corrections from a English native speaker should be carried out
Round 2
Reviewer 3 Report
Comments and Suggestions for Authors
The authors have successfully addressed all the comments and improved significantly the quality of the paper